# Kemeny-based testing for COVID-19

**Serife Yilmaz[1], Ekaterina Dudkina[2], Michelangelo Bin[3], Emanuele Crisostomi[2]\*,
Pietro Ferraro[1], Roderick Murray-Smith[4], Thomas Parisini[3,5,6], Lewi Stone[7],
Robert Shorten[1]**

**1** Dyson School of Design Engineering, Imperial College London, London, United Kingdom, **2** Department of Energy, Systems, Territory and Constructions Engineering, University of Pisa, Pisa, Italy, **3** Department of Electrical and Electronic Engineering, Imperial College London, London, United Kingdom, **4** School of Computing Science, University of Glasgow, Glasgow, Scotland, **5** Department of Engineering and Architecture, University of Trieste, Trieste, Italy, **6** KIOS Research and Innovation Center of Excellence, University of Cyprus, Nicosia, Cyprus, **7** The George S. Wise Faculty of Life Sciences, Tel Aviv University, Tel Aviv-Tafo, Israel

\* emanuele.crisostomi@unipi.it

**Data Availability Statement:** All relevant data are within the paper.

**Funding:** Funding for this study was provided by the following organizations in the form of grants: the Engineering and Physical Sciences Research

## Abstract

Testing, tracking and tracing abilities have been identified as pivotal in helping countries to safely reopen activities after the first wave of the COVID-19 virus. Contact tracing apps give the unprecedented possibility to reconstruct graphs of daily contacts, so the question is: who should be tested? As human contact networks are known to exhibit community structure, in this paper we show that the Kemeny constant of a graph can be used to identify and analyze bridges between communities in a graph. Our 'Kemeny indicator' is the value of the Kemeny constant in the new graph that is obtained when a node is removed from the original graph. We show that testing individuals who are associated with large values of the Kemeny indicator can help in efficiently intercepting new virus outbreaks, when they are still in their early stage. Extensive simulations provide promising results in early identification and in blocking the possible 'super-spreaders' links that transmit disease between different communities.

## Introduction

### Motivation

Amidst fears of a possible second wave of the COVID-19 disease, methodologies based around *test, track, and trace* (3T) policies have been identified in many countries to lift confinement restrictions [1–3]. Some initial examples of contact tracing methods, that have been widely applied from the beginning of the epidemic, aimed at combining data from interviews, smartphones' GPS and Bluetooth histories, credit cards and camera records. Examples of successful applications in China, Iceland, New Zealand, Singapore, South Korea and Taiwan, have been shown to contribute to mitigating the spread of the virus [4], but such *manual* contact tracing approaches are particularly time consuming, especially with large numbers of infected people. Besides, considering the long incubation period, and exponential growth in transmission, even short delays in actions may lead to the loss of control of the epidemic [5]. From this

Council (EPSRC), grant EP/R018634/1, Closed-loop Data Science, awarded to RMS; the European Union's Horizon 2020 Research and Innovation Programme, Grant Agreement No 739551 (KIOS CoE), awarded to MB and TP; Science Foundation Ireland, grant 16/IA/4610, awarded to RS; and the EPSRC, grant EP/V018450/1, awarded to RS, RMS, and TP.

**Competing interests:** The authors have declared that no competing interests exist.

perspective, it has also been shown in [6] that reducing the delay in detecting a new case from 5 to 3 days leads to a 60-70% improvement in efficiency, in terms of reproduction number. Accordingly, an alternative more efficient solution for contact tracing includes the use of mobile phone applications with immediate notification. Considering a high penetration rate, and a high compliance of people in using this app, it could significantly help to stop the epidemic as shown in [5]. The benefits of efficient testing are clear. In addition to identifying infected individuals and tracing their contacts, fast diagnostic tests also allow estimation of the degree of spread of the virus in a region.

Accordingly, one proposal is to perform the tracing task by using Bluetooth connectivity to recognize when a prolonged proximity between two smartphones (and thus, their owners) occurs. For instance, the smartphone app that has been recommended by the Italian government stores a contact when a proximity of less or equal than two meters for at least 15 minutes is recorded [7]. Thus, the tracing task is currently designed as a *reactive* process, as it is a reaction to a positive test. Consequently, unless the positive individual is tested in the very early stages of the infection, which may be unrealistic in practice in many cases since symptoms do not usually appear before 3-4 days, large numbers of naive tests and ineffective quarantines may be required for an effective 3T policy. It is therefore of interest to reshape the sampling process to be proactive and make sampling more efficient in the presence of limited testing capabilities. In this context, an obvious related questions arises: *Is it possible to identify the potential super-spreaders before they spread?* [1]. While such a question implies a formal definition of what is a 'super-spreader', we note that in the context of COVID-19, the effect of the disease is highly compartmentalized; not only regionally, but also in terms of demographics, with older communities highly at risk, and younger children apparently in a low risk category, but still with the potential for acting as vectors. Indeed, the terrible effects of the disease in care homes for the elderly, and in communities such as the Satmar community [8] only reinforce the negative consequences of the disease jumping from one compartment to another. Our main objective is therefore to: *Propose testing strategies which can identify 'bridges' between communities, which could easily become 'super-spreaders'*. For this purpose, we shall describe how the contact tracing apps can be conveniently used to support targeted prioritization of testing, by exploiting the reconstructed networks of daily contacts. Despite the fact that current apps are not explicitly designed for this purpose, as they are only used when an individual is identified as being infected, we show that the network information they are capable of acquiring could actually provide very valuable information also in the absence of infected individuals.

## State of the art

The issue of *who should be tested* is not new in epidemiology. This problem is very similar to the classic one of who should be vaccinated in a population. For instance, it is well established that random immunization requires immunizing a very large fraction of a population in order to abate contact-transmissible epidemics [9]. When time or resources are limited, better results can be achieved if smarter immunisation strategies are used; see, for instance the *immunising random acquaintances of random nodes* policy [9] which is known to be more successful than a fully random strategy in identifying the super-spreaders.

In principle, targeted immunization of the most highly connected individuals is also known to be more effective. However, since such vaccination policies actually require a global knowledge of the contact network, they are impractical in most cases [9]. In the COVID-19 context, the advantage of sample testing is confirmed in [10], where the authors analyzed the effect of testing, isolation, tracing, physical distancing, and type of contacts (household and others) on

the reproduction number. According to their simulations, focused testing strategies based on contact tracking help to hold back the epidemic more efficiently than widespread mass testing or self-isolation alone. From this perspective, it is clear that the contact tracing apps provide an unprecedented opportunity to infer the network over which an epidemic spreads, and to implement targeted immunization/testing policies [2].

Assuming that the tracing apps truly do provide a snap-shot of a city-wide network of contacts, then the problem becomes: what is meant by the most highly connected individuals? Such problems are highly topical in computer science, mathematics, and engineering, and a number of tools are available to us. In this paper we compare several measures. We first consider *the graph node degree* (i.e., the number of daily contacts) as the most obvious option of who should be tested. An alternative is represented by the Google's PageRank indicator [11], as suggested in [12]. These indicators identify influential contacts in a graph, and do not necessarily identify communities or individuals that bridge communities, and so, as we shall show, may not be particularly effective for our purpose. Thus, we also consider an indicator based on the Kemeny constant [13] which we shall show to be particularly attractive for this task.

## Materials and methods

### A primer on Markov chains

Graph theory and Markov chains have been ubiquitously employed in many different fields of engineering and applied mathematics, including epidemiology [14, 15]. Here we only briefly recall some basic notions that will be later used in our analysis. In doing this, we follow [13], based on classic references [11, 16].

In this manuscript, we shall only consider discrete-time, finite-state, homogeneous Markov chains. In this situation, the Markov chain is a discrete time stochastic process $x_k, k \in \mathbb{N}$, and characterized by the equation

$$p(x_{k+1} = S_{i_{k+1}} | x_k = S_{i_k}, ..., x_0 = S_{i_0}) = p(x_{k+1} = S_{i_{k+1}} | x_k = S_{i_k}) \ \forall k \geq 0, \tag{1}$$

where $p(E|F)$ denotes the conditional probability that event $E$ occurs given that event $F$ occurs.

A Markov chain with $n$ states is completely described by the $n \times n$ transition probability matrix $\mathbb{P}$, whose entry $\mathbb{P}_{ij}$ denotes the probability of passing from state $S_i$ to state $S_j$ in exactly one step. $\mathbb{P}$ is a row-stochastic non-negative matrix, as the elements in each row are probabilities and they sum up to 1. Within Markov chain theory, there is a close relationship between the transition matrix $\mathbb{P}$ and a corresponding graph. The graph consists of a set of nodes that are connected through edges. The graph associated with the matrix $\mathbb{P}$ is a directed graph, whose nodes are given by the states $S_i$, $i = 1, \ldots, n$, and there is a directed edge leading from $S_i$ to $S_j$ if and only if $\mathbb{P}_{ij} \neq 0$. A graph is strongly connected if for each pair of nodes there is a sequence of directed edges leading from the first node to the second one. The matrix $\mathbb{P}$ is irreducible if and only if its directed graph is strongly connected. Some important properties of irreducible transition matrices follow from the well-known Perron-Frobenius theorem [11]:

- The spectral radius of $\mathbb{P}$ is 1; in particular, 1 belongs to the spectrum of $\mathbb{P}$, and has an algebraic multiplicity of 1;

- The left-hand Perron eigenvector $\pi$ is the unique vector defined by $\pi^T \mathbb{P} = \pi^T$, such that every single entry of $\pi$ is strictly positive and $\|\pi\|_1 = 1$. Except for positive multiples of $\pi$ there are no other non-negative left eigenvectors for $\mathbb{P}$.

One of the main properties of irreducible Markov chains is that the $i'$th component $\pi_i$ of the vector $\pi$ represents the long-run fraction of time that the chain will be in state $S_i$. The row vector $\pi^T$ is also called the stationary distribution vector of the Markov chain.

In our application, a node of the graph is an individual with the contact tracing app installed in her/his smartphone. Two nodes are connected through an undirected edge if the app recognizes that two individuals have been in close contact for a sufficient time (e.g., 15 minutes in the previous example of the Italian app). If the app also records the amount of time, or the distance, between two individuals, then it would be possible to consider a weighted graph, where the weights could correspond to the probability of contagion (i.e., it would increase with the duration of the contact, and decrease with the distance), as will be considered in a specific later section. Also, note that the contact tracing apps give rise, in principle, to daily graphs which are not fully connected. This is due to the fact that some communities may be in fact isolated, and in general single individuals may not have significant contacts with other people during a day.

## Mean first passage times and the Kemeny constant

A transition matrix $\mathbb{P}$ with 1 as a simple eigenvalue gives rise to a singular matrix $I - \mathbb{P}$ (where the identity matrix $I$ has appropriate dimensions), which is known to have a group inverse $(I - \mathbb{P})^{\#}$. The group inverse is the unique matrix such that $(I - \mathbb{P})(I - \mathbb{P})^{\#} = (I - \mathbb{P})^{\#}(I - \mathbb{P})$, $(I - \mathbb{P})(I - \mathbb{P})^{\#}(I - \mathbb{P}) = (I - \mathbb{P})$, and $(I - \mathbb{P})^{\#}(I - \mathbb{P})(I - \mathbb{P})^{\#} = (I - \mathbb{P})^{\#}$. More properties of group inverses and their applications to Markov chains can be found in [16]. The group inverse $(I - \mathbb{P})^{\#}$ contains important information on the Markov chain and it will be often used in this paper. For this reason, it is convenient to denote this matrix as $Q$. The mean first passage time (MFPT) $m_{ij}$ from the state $S_i$ to the state $S_j$ denotes the expected number of steps to arrive at destination $S_j$ when the origin is $S_i$, and the expectation is averaged over all possible paths following a random walk from $S_i$ to $S_j$. If we denote by $q_{Ij}^{\#}$ the $ij$ entry of the matrix $Q$, then the mean first passage times can be computed according to [17],

$$m_{ij} = \frac{q_{jj}^{\#} - q_{ij}^{\#}}{\pi_j} \qquad i, j = 1, ..., n, i \neq j. \tag{2}$$

We assume that $m_{ii} = 0$. The Kemeny constant is defined as

$$K = \sum_{j=1}^{n} m_{ij}\pi_j, \tag{3}$$

where the right-hand side is independent of the choice of the origin state $S_i$ [16]. An interpretation of this result is that the expected time to get from an initial state $S_i$ to a destination state $S_j$ (selected randomly according to the stationary distribution $\pi$) does not depend on the starting point $S_i$ [18]. Therefore, the Kemeny constant is an intrinsic measure of a Markov chain, and if the transition matrix $\mathbb{P}$ has eigenvalues $\lambda_1 = 1, \lambda_2, ..., \lambda_n$, then another way of computing $K$ is [19],

$$K = \sum_{j=2}^{n} \frac{1}{1 - \lambda_j}. \tag{4}$$

As can be seen from Eq 4, $K$ is only related to the particular matrix $\mathbb{P}$ and it becomes very large if one or more of the other eigenvalues of $\mathbb{P}$, different from $\lambda_1$, are close to 1.

**Remark**. The Kemeny constant admits many interpretations. First, it is related to the mean first passage times of the underlying Markov chain. But it is much more than this. It is also

determined by the *entire* spectrum of the transition matrix. From a control theoretic perspective it resembles the sum of rise times along all the modes of the system. Thus, while the second eigenvalue of the transition matrix gives a bound on the convergence rate of the underlying Markov chain, the Kemeny constant is akin to an average of rise times across all modes.

The Kemeny constant is usually computed using the group inverse (Eq 3) or the knowledge of all the eigenvalues (Eq 4). A more convenient computation, revealing the complexity of the calculation, can be developed as follows. Let $\mathbb{P}$ be a $n \times n$ stochastic, irreducible transition matrix. We denote the eigenvalues of $\mathbb{P}$ by $\lambda_1 = 1, \lambda_2, \ldots, \lambda_n$ and its characteristic polynomial is

$$p(s) = \det(sI - \mathbb{P}) = (s-1)(s-\lambda_2)\cdots(s-\lambda_n).$$

We define $\tilde{p}(s) = p(s)/(s-1)$. The Kemeny constant of $\mathbb{P}$ can be calculated using its characteristic polynomial:

$$
\begin{aligned}
K &= \frac{1}{1-\lambda_2} + \frac{1}{1-\lambda_3} + \cdots + \frac{1}{1-\lambda_n} \\
&= \frac{\tilde{p}'(1)}{\tilde{p}(1)}.
\end{aligned}
$$

Since $p(s) = (s-1)\tilde{p}(s)$,

$$\tilde{p}(1) = \lim_{s \to 1} \frac{p(s)}{s-1} = \lim_{s \to 1} \frac{p'(s)}{1} = p'(1)$$

and using the derivative of $\tilde{p}(s) = p(s)/(s-1)$ we get

$$
\begin{aligned}
\tilde{p}'(1) &= \lim_{s \to 1} \frac{p'(s).(s-1) - p(s)}{(s-1)^2} \\
&= \lim_{s \to 1} \frac{p''(s).(s-1) + p'(s) - p'(s)}{2(s-1)} \\
&= \frac{1}{2} p''(1).
\end{aligned}
$$

Hence, the Kemeny constant of the matrix $\mathbb{P}$ can be written as

$$K = \frac{1}{2} \frac{p''(1)}{p'(1)}. \tag{5}$$

**Remark**.

- A characteristic polynomial interpretation of $K$ is also given in [20, 21]. To the best of our knowledge the link to the characteristic polynomial and the Markov transition matrix, more precisely $I - \mathbb{P}$, was first given in [20]. A further derivation is given in [21], this time using the associated adjacency matrix. However, the derivation given here expresses $K$ from the derivative of the characteristic polynomial associated with $\mathbb{P}$.

- Calculation of the Kemeny constant is of the same computational complexity as that of calculating the determinant.

- The determinantal interpretation of $K$ suggests a deeper control theoretic interpretation of the Kemeny constant.

## Kemeny indicator for criticality

A generic criticality measure of the $i'$th node in a graph can be computed as the global critical-ity measure of the graph after the removal of node $i$ [22], we quantify the importance of each single node as the corresponding value of the Kemeny constant of the graph obtained from the original graph by removing such a node, and for simplicity we shall denote it as the Kemeny indicator. The rationale of this choice is that nodes connecting two communities are associated with very high Kemeny indicators, as walking times become much larger if such nodes are removed. In particular, if a single node connects two communities then it implies that only one person belongs to both such communities, and if that node is removed, then the Kemeny constant tends to infinity (i.e., after removing that node, the graph is split into two non-con-nected sub-graphs, and it is impossible to find a path from one community to the other com-munity, thus the walking time tends to infinity).

## PageRank and betweenness centrality indicators

As a starting point, we are first interested in undirected and un-weighted graphs. In this case, if teleportation is not considered, then PageRank simply corresponds to the Perron eigenvec-tor, and it is well-known that node degree and the Perron eigenvector are highly correlated [23]. Here, we remind that teleportation is an action where one stops following the Web's hyperlink structure and jumps to a totally new page at random. This for instance occurs imme-diately after entering a dangling node, i.e., a webpage with no outlinks [11]. In addition, our graph structures have some other important properties. In particular, if we denote by $A$ the symmetric [0, 1] adjacency matrix that has ones in positions $A_{i,j}$ and $A_{j,i}$ if individuals $i$ and $j$ are in close contact for a long enough time, then it can be noticed that the row-stochastic matrix $\mathbb{P}$ of our interest can be obtained as

$$\mathbb{P} = D^{-1}A, \qquad (6)$$

where $D$ is the diagonal matrix, whose $D_{ii}$ entry corresponds to the degree of the $i$'th node of $A$. Also, the eigenvalues of the row-stochastic (non-symmetric) matrix $\mathbb{P}$ are the same of the symmetric matrix $D^{-1/2}AD^{-1/2}$, and we remind the reader that the eigenvalues of symmetric matrices are real. Thus: all eigenvalues and eigenvectors of $\mathbb{P}$ are real as well. This shall play an important role in the following discussion.

Another measure of centrality in graphs is represented by betweenness centrality. In its basic definition, it measures the number of shortest paths that pass through a node [24, 25]. In principle, it is known to show which nodes are acting as "bridges" between communities in graphs.

However, in many networks, including the contact networks of our interest, the informa-tion (or here, the virus) does not flow along shortest paths, and will most likely take a random route [26]. Accordingly, a measure of betweenness centrality based on random walks, called *Random Walk Betweenness (RWB)* was introduced in [26]. This measure was shown to better rank the importance of nodes in graphs with existing communities, and to be less correlated with vertex degree in most networks [26]. Also, it is known that in networks with strong com-munity structure, immunization interventions targeted at individuals bridging communities (e.g., using random walk betweenness) are more effective than those simply targeting highly connected individuals [27]. As we shall see, RWB appears very effective in identifying candi-date nodes to be tested, and its main drawbacks appear to be the intensive computational bur-den required to compute it, and its definition that can be directly applied only to undirected and un-weighted graphs.

**Remark**. The Kemeny constant may be interpreted as the average time to take a random walk in the contagion graph, weighted according to likely destinations. As such it takes into account the stationary distribution, and the first mean passage times from a given starting location and all other destinations. Accordingly, this constant represents a compromise between indicators that use only the stationary distribution (such as PageRank and node degrees), and those using path-based algorithms (such as the betweenness centrality indicators). Thus, the indicator should work in highly connected single-community graphs, and in more sparse graphs, associated with many unknown sub-communities.

## Results

### Who should be tested: Node degree vs. Kemeny indicator

We use the Kemeny-based indicator as a proxy to determine individuals that should be tested.

We now consider a simple scenario with a population of 240 individuals who belong to 6 different communities (with 40 people within each community). Communities are artificially created by assuming that there is high probability that there exists an edge that connects individuals belonging to the same community (19%), and a low probability that there exists an edge that connects individuals belonging to different communities (0.1%). With such values, a modularity of 0.8 is obtained which is consistent with known communities [27]. In the assumption that 10 tests were available, Fig 1 shows the 10 individuals with highest degree (magenta) and the 10 individuals with highest Kemeny indicator (black). By visual inspection, it is very simple from Fig 1 to understand the main difference between Kemeny-based testing, and testing the individuals with highest node degree. In particular, the Kemeny indicator identifies the bridges between the communities, i.e., people that visit different communities during the same day, regardless of how many people in total they meet during the day. While, in

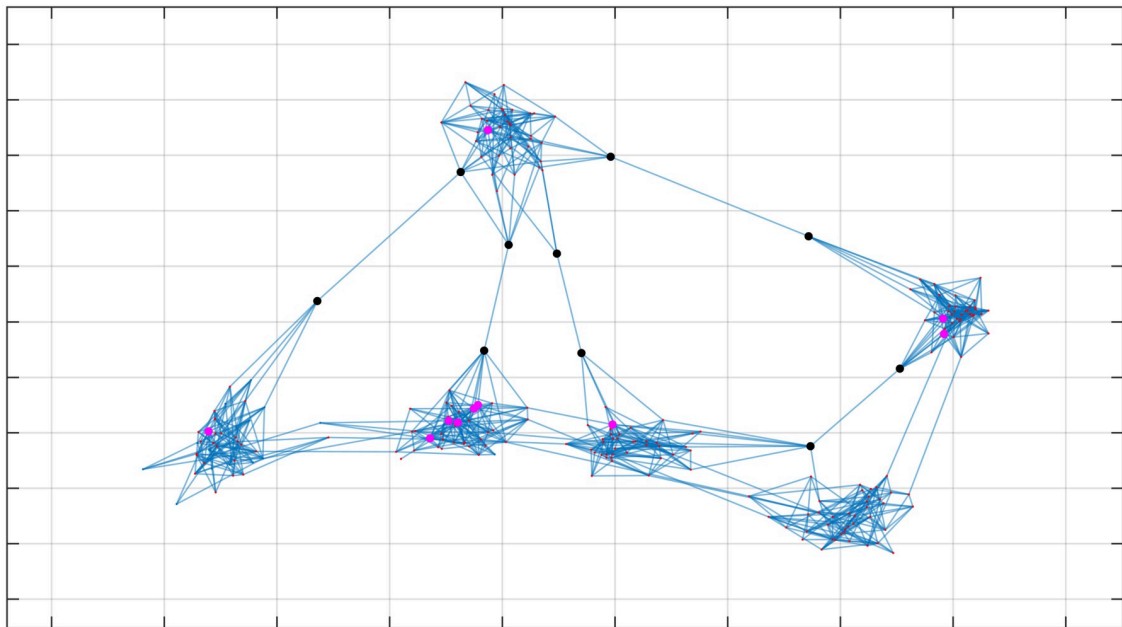

**Fig 1. Kemeny indicator vs. Node degree.** The six existing communities can be clearly identified by visual inspection. Assuming that up to 10 individuals can be tested, the black circles show those that would be chosen according to their highest Kemeny value, while the nodes in magenta correspond to the nodes (i.e., individuals) with highest degree. Edges correspond to the random interactions on one day.

principle, benefits may be found for both testing people with high node degree or high Kemeny indicator, we argue that this feature of the Kemeny indicator may be particularly convenient as a better coverage of the graph is obtained (the same individual covers more than only one community), and also is convenient to intercept possible new virus outbreaks occurring in communities, when they are still in their early stages.

**Remark**. For simplicity, in this paper we always compute in a single shot all nodes that rank in the first $n$ positions, assuming $n$ tests are available, independently from the method/indicator we choose for ranking. As an alternative, one could compute the node that ranks first, and then remove it from the network, and compute the node that ranks first in the new network, and iteratively repeat the procedure $n$ times to compute the top $n$ nodes.

## Impact of communities on indicators

The previous case study, exemplified in Fig 1, assumed the presence of communities. We now show what happens when a strong community structure does not exist. For this purpose, Figs 2 and 3 compare how different indicators provide different outcomes depending on the existence or not of communities. In particular, Fig 2 refers to a case where people meet with the same probability (19% again) people belonging to their community and to other communities (i.e., communities degenerate into a single large community). In this case, Fig 2 shows that both indicators based on node degrees (on the left-hand side) and those based on random walks (our proposed Kemeny indicator and RWB, on the right-hand side) provide the same results. On the other hand, when the probability of meeting people of other communities is decreased, then communities emerge from the graph, and the two categories of indexes clearly provide different results, see Fig 3.

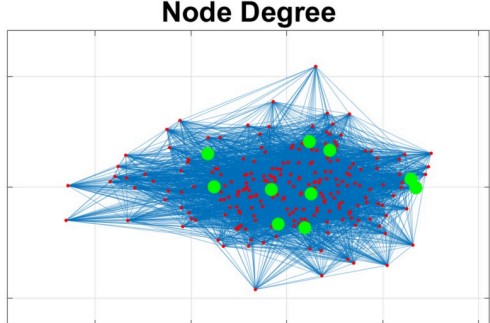

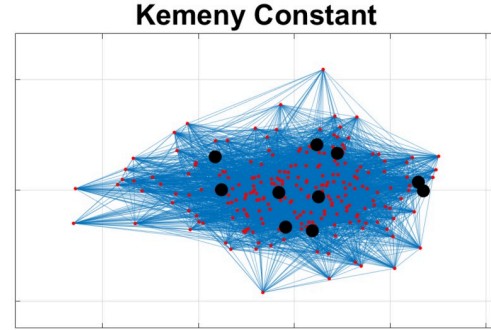

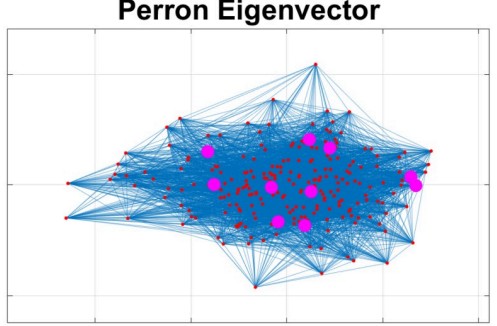

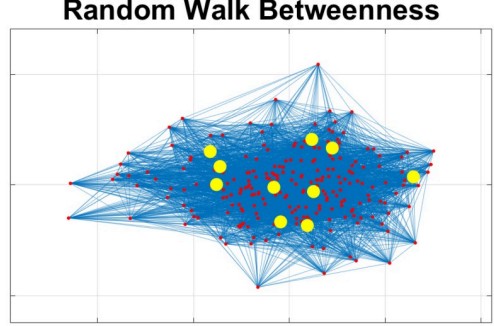

**Fig 2. Comparison of different indicators in networks without community structure.** If no communities exist, then all indicators appear to select the same nodes (i.e., those with highest degree) for testing.

### Node Degree

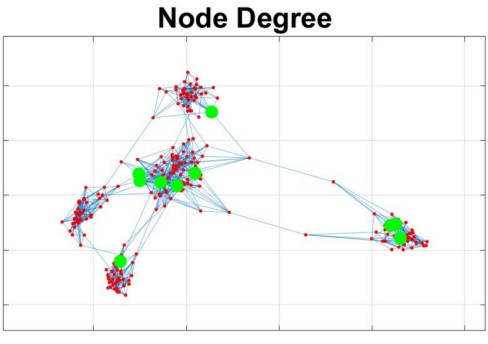

### Kemeny Constant

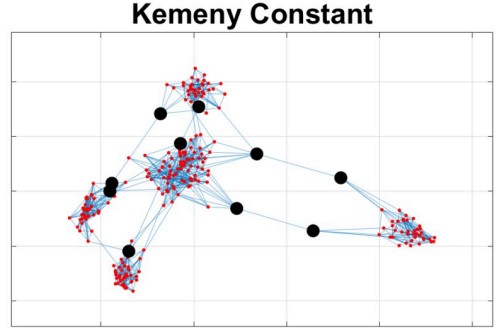

### Perron Eigenvector

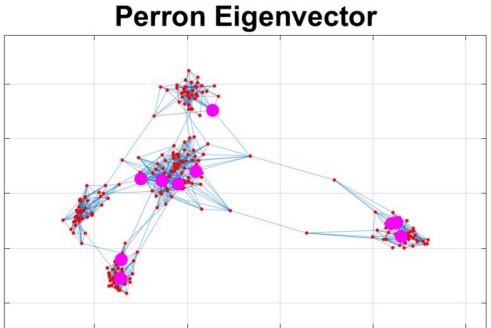

### Random Walk Betweenness

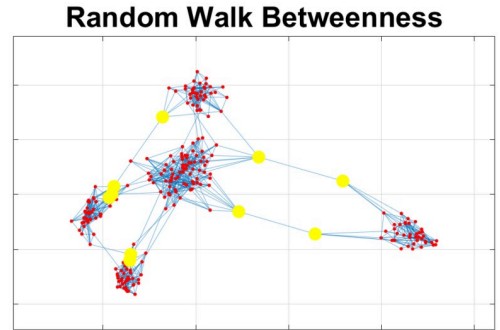

**Fig 3. Comparison of different indicators in networks with community structure.** When communities arise, then it is easier to appreciate the different strategies pursued by indicators based on node degrees (left) and those based on random walks (right).

## Impact of tests on the dynamics of an epidemics

We now try to quantitatively evaluate the impact of using one specific indicator over another one (i.e., Kemeny indicators vs. Node degree) in terms of the spreading of the virus. For this purpose, every day we consider a graph created according to the usual probabilities previously outlined. On the first day, we assume that 2 individuals randomly chosen are infected. Then, every day, each susceptible individual who enters in contact with an infected individual has a probability 10% of being infected. In addition, every day, using the information retrieved by the contact tracing app, we assume that the individuals who rank in the top positions according to different indicators are tested. If they are found infected, then they are quarantined for two consecutive weeks. Also, in this case, all their contacts of the same day are tested (and quarantined if positive). Note that in principle the procedure may be iterated in the past (i.e., the contacts of the previous days may be tested as well), but this is not considered here for simplicity. Such a simple case study may be associated with the spreading of the virus in a network of asymptomatic infected individuals, as we do not consider individuals who may autonomously decide to get tested (e.g., because they have developed symptoms), and only individuals detected by the test are quarantined.

Fig 4 shows the results of running 20 repetitions of the simulations for 30 days (repetitions are used to obtained averaged results). The Kemeny indicator gives an improvement by reducing the number of people infected, with a comparable total number of days spent in quarantine by the population. With the settings used, when more than 10 tests are assumed to be available, then with the Kemeny indicator, it becomes much more likely to restrict the infection to the local communities initially infected. Also, it is possible to appreciate that much better results are obtained by testing only 20 individuals per day, targeting the individuals with highest

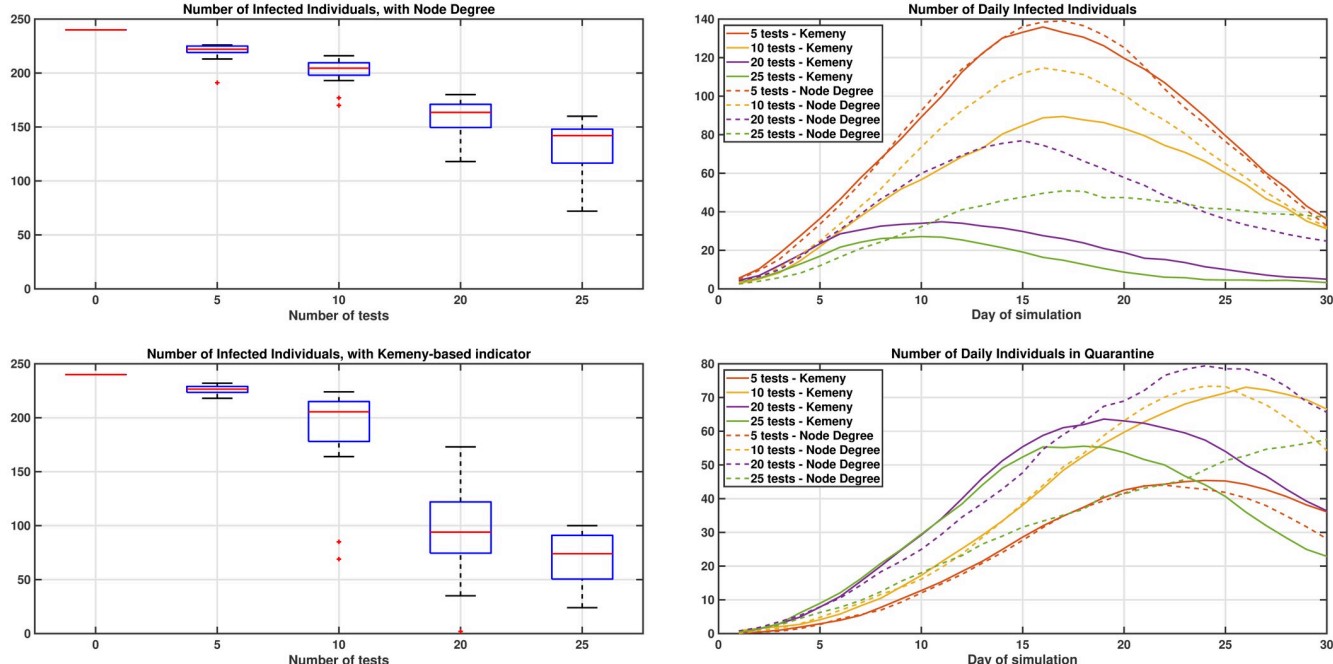

**Fig 4. Dynamics of the virus for different testing strategies.** Impact of testing individuals using metrics based on the Node Degree (top) or Kemeny indicator (bottom) on the left-hand side. On the right-hand side, the average number of infected and quarantined individuals on each day of the simulation. Figures are averaged over 20 repetitions of the simulation process, to give insight into the level of variability between runs.

Kemeny indicator, than testing 25 individuals per day, targeting those with the highest node degree. In Fig 4, on the right hand side, it can be also seen that the number of infected individuals decreases steadily when more people per day are tested, while the number of individuals in quarantine oscillates (i.e., it is low when very few individuals are tested, because few tests are used, and when a lot of people are tested as well, because fewer individuals are infected and require quarantine; highest values are achieved for intermediate numbers of tests).

Fig 5 shows the results of running 20 repetitions of the simulations for 30 days but with a fixed random network. The network is sampled with the same strategy before on the first day, but then kept fixed for further days. All other aspects of the simulation are as before. In this case, a paradoxical result is obtained, which is that the same individuals are tested, most likely, every day (because most part of the graph remains fixed, and accordingly, more likely the same individuals have highest node degree, or give rise to the highest values of the Kemeny indicator). Accordingly, much fewer individuals than before are quarantined during the simulations.

## Simulations on benchmark graphs

The objective of this section is to validate the previously described findings in networks that are known to realistically capture relations in contact networks. In particular, we follow the procedure described in [27] (where it was observed that human contact networks exhibit strong community structure, in the context of immunization interventions) to generate networks with community structure, which we briefly report below, taken from [27] (for simplicity, sizes have been scaled to be consistent with our simulations, and to allow for simple visual inspections of the results):

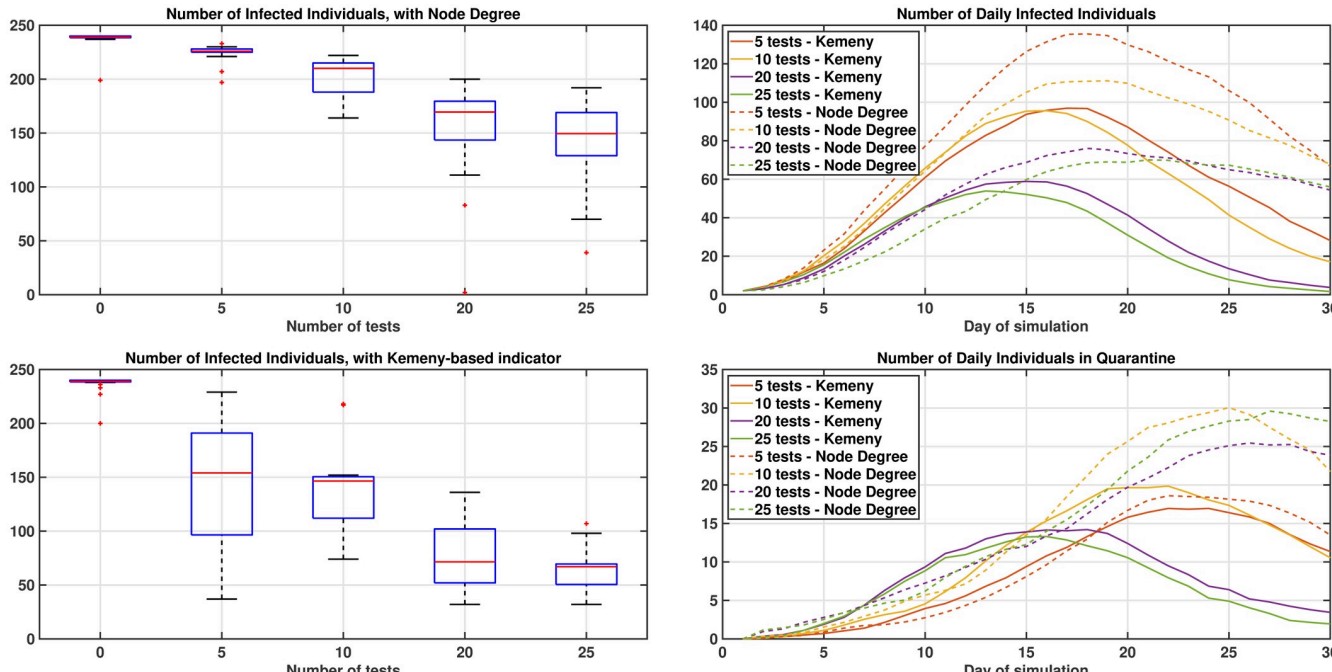

**Fig 5. Dynamics of the virus for different testing strategies, with a fixed graph.** Impact of testing individuals on a fixed network using metrics based on the Node Degree (top) or Kemeny indicator (bottom) on the left-hand side. On the right-hand side, the average number of infected and quarantined individuals on each day of the simulation. Figures are averaged over 20 repetitions of the simulation process, and on average fewer individuals are quarantined in this case, as most likely the same individuals are tested every day.

1. 6 small-world communities of 40 nodes are first created using the Watts-Strogatz algorithm [28], so that each node has exactly 8 edges connecting to nodes of the same community;

2. We then add 240 edges randomly to connect different communities;

3. We then rewire *between-communities* edges so that they become *within-community* edges. In doing this, the modularity of the graph increases, and we stop the procedure once a desired level of modularity is achieved.

In particular, we terminate the previous procedure when a modularity equal to 0.8 is achieved (as before), as this is known to be consistent with many contact networks investigated in the literature [27].

Also for such realistic networks, very similar results have been obtained, as depicted in Fig 6.

The adopted procedure gives rise to networks with an average degree of 10, which is in line with general findings regarding social contact patterns. However, it is reasonable to presume that after a first wave of COVID-19, individuals will meet fewer people than they did before the disease (due to adoption of non-pharmaceutical interventions, like observation of distance measures). Accordingly, Fig 7 further compares Kemeny-testing and node degree testing, assuming that the initial communities are created with node degree 8 (as before), 6 and 4 (step 1 of the previous procedure; final modularity is maintained constant equal to 0.8). When networks with smaller average node degrees are considered, then fewer edges appear in the graph (including fewer edges that connect different communities), and the improvement of Kemeny-based testing over node degree becomes more relevant than before.

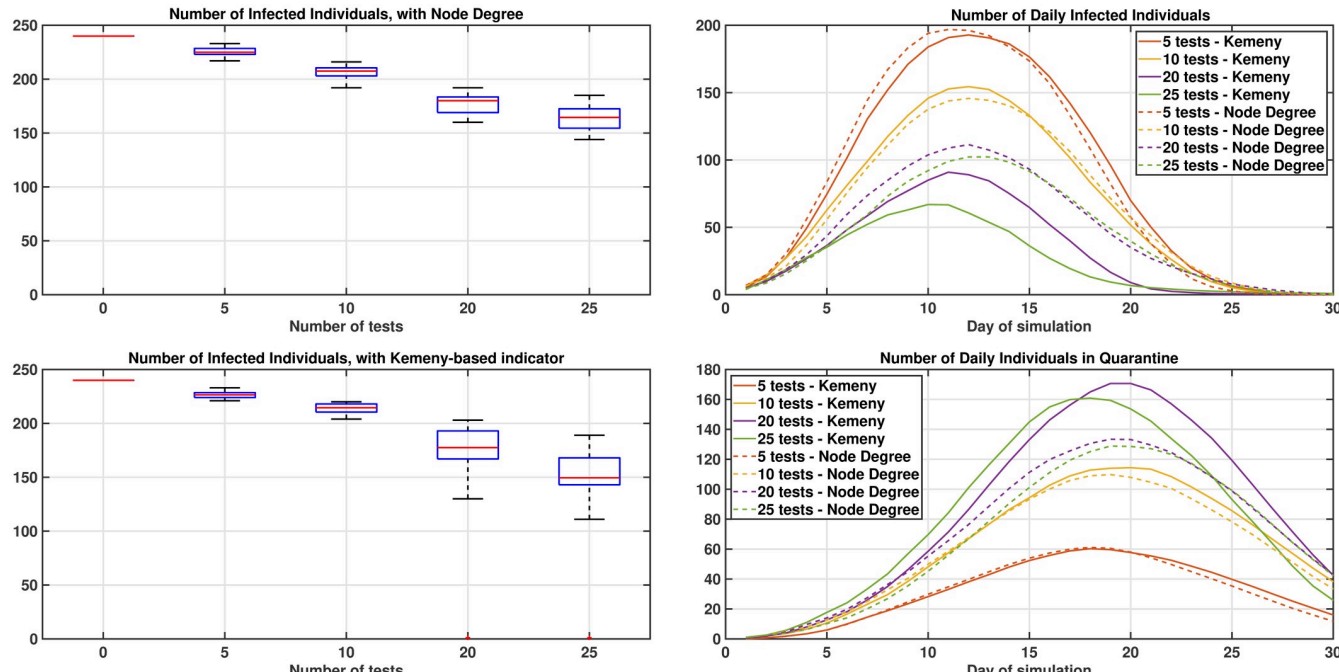

**Fig 6. Comparison in benchmark graphs.** Impact of testing individuals using metrics based on the Node Degree (top) or Kemeny indicator (bottom), in a realistic graph created according to the procedure proposed in [27]. The left-hand figures show the total number of people who were infected during a 30-day simulation. The right hand figures show the cumulative number of days spent in quarantine for members of the population (N = 240). The boxplots show the distribution of 20 repetitions of the simulation process, to give insight into the level of variability between runs.

## Effect of compliance

An important issue related to contact tracing apps, regards whether people will indeed comply with the national, or local, recommendations of downloading and installing the same app to reconstruct daily graphs of contacts [4]. This issue appears particularly compelling for the case of Kemeny-based testing: roughly speaking, Kemeny-based testing works well because bridges between communities are identified, which implies that individuals travelling from one (already infected) community to another (fully healthy) community are detected in the early stages of a new virus outbreak. However, if only a fraction of the population runs the contact tracing app, then some community bridging individuals may be missed, undermining the main strengths of Kemeny indicator-based tests. On the other side, it is less clear how compliance would affect node degree-based testings.

For this purpose, Fig 8 compares the percentage of individuals who remain healthy throughout the 30-day simulation assuming that 20 individuals (out of a population of 240) are tested every day according to either the Kemeny-based indicator or according to the Node Degree (results are averaged over 20 different simulations, to filter out stochastic effects). It is interesting to note that independently from the fraction of individuals who install the app, and thus are traced in practice, the Kemeny-based indicator appears to consistently outperform the one based on the node degree.

## Weighted undirected graphs

So far, we have restricted our interest to unweighted undirected graphs. One reason for doing so is that most centrality indicators are usually defined upon the adjacency matrix (e.g., node degree or random walk betweenness), rather than on the transition matrix. However,

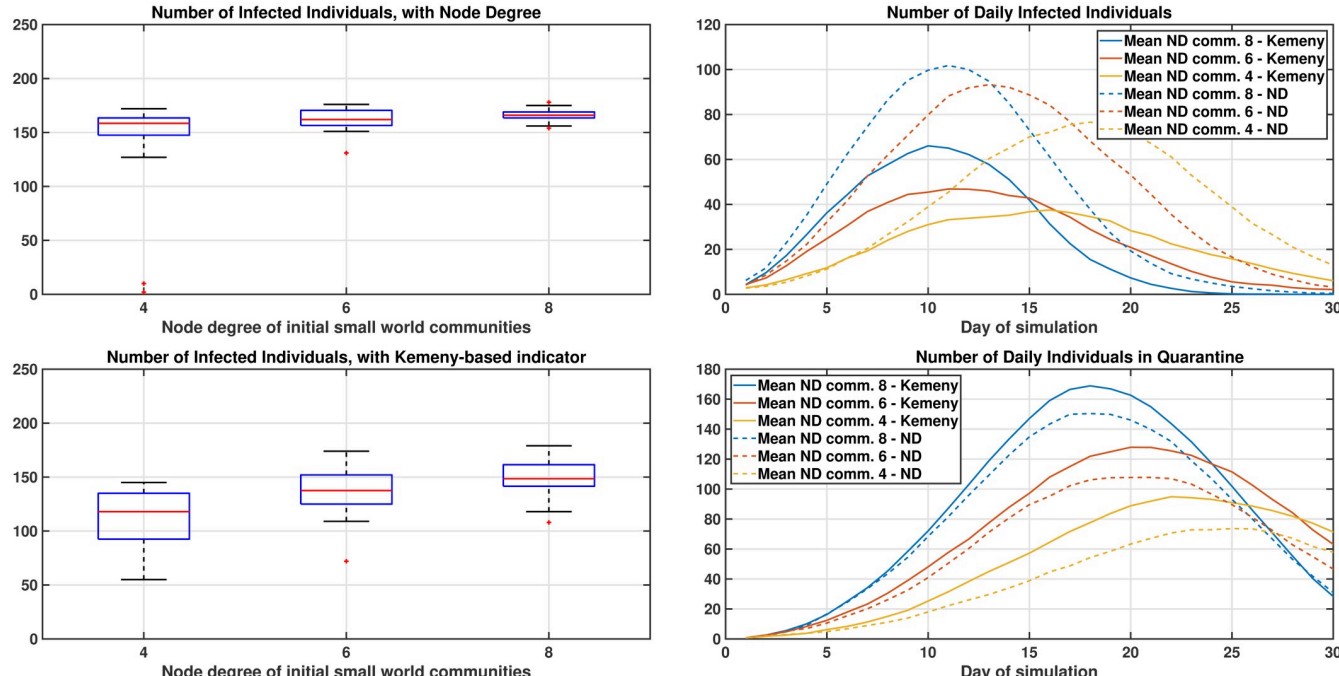

**Fig 7. Comparison in benchmark graphs for graphs with different average node degree.** Impact of testing individuals using metrics based on the Node Degree (top) or Kemeny indicator (bottom), in a realistic graph created according to the procedure proposed in [27], for graphs with different average node degree is shown on the left-hand side. While the community structure of the graph remains the same, when smaller node degree are considered, the improvement of Kemeny-based testing over node degree testing becomes more relevant. 25 daily tests, and 20 repetitions of each instance have been considered. On the right-hand side, the number of infected and quarantined individuals is shown.

assuming that the contact tracing app does not simply store the information about whether a contact has occurred or not, but also the duration of a contact, or the distance between two individuals during a contact, then a different probability of infection may be associated with each different contact.

We now provide a simple solution to take into account weighted graphs, where we assume that the weights are proportional to the duration of a contact. In particular, we assume that a probability equal to 1 is associated with a contact that lasts for 10 hours (and also for durations longer than 10 hours), and a probability equal to 0.025 (i.e., 2.5%) for contacts that last 15 minutes. Contacts that last less than 15 minutes are not recorder by the app, while for contacts of intermediate duration between 15 minutes and 10 hours, the probability changes in a linear fashion (in practice, for such intermediate values, the probability is equal to the duration of the contact in minutes divided by 600). While this is a very simple way to model the probability of infection proportionally to the duration of a contact, any other more sophisticated model may be used without affecting our general discussion at all.

In addition, we add an extra 'idle' state to represent the possibility that one infected individual does not pass the infection to anybody else (e.g., because he/she had no contacts during the day). Then, from the idle state, we assume that with equal probability it is possible to pass to any other state (i.e., to any individual in the network). The trick of adding an extra idle state has been well explored in the Markov chain community, for instance to avoid having absorbing states in the chain, and sometimes is denoted as "teleportation" [11]. In our application, the interpretation of the teleportation possibility is particularly convenient and meaningful, as it corresponds to the possibility that the virus is spread without a proper contact occurring: so this can be used to model the fact that individuals may get infected even after contacts that last

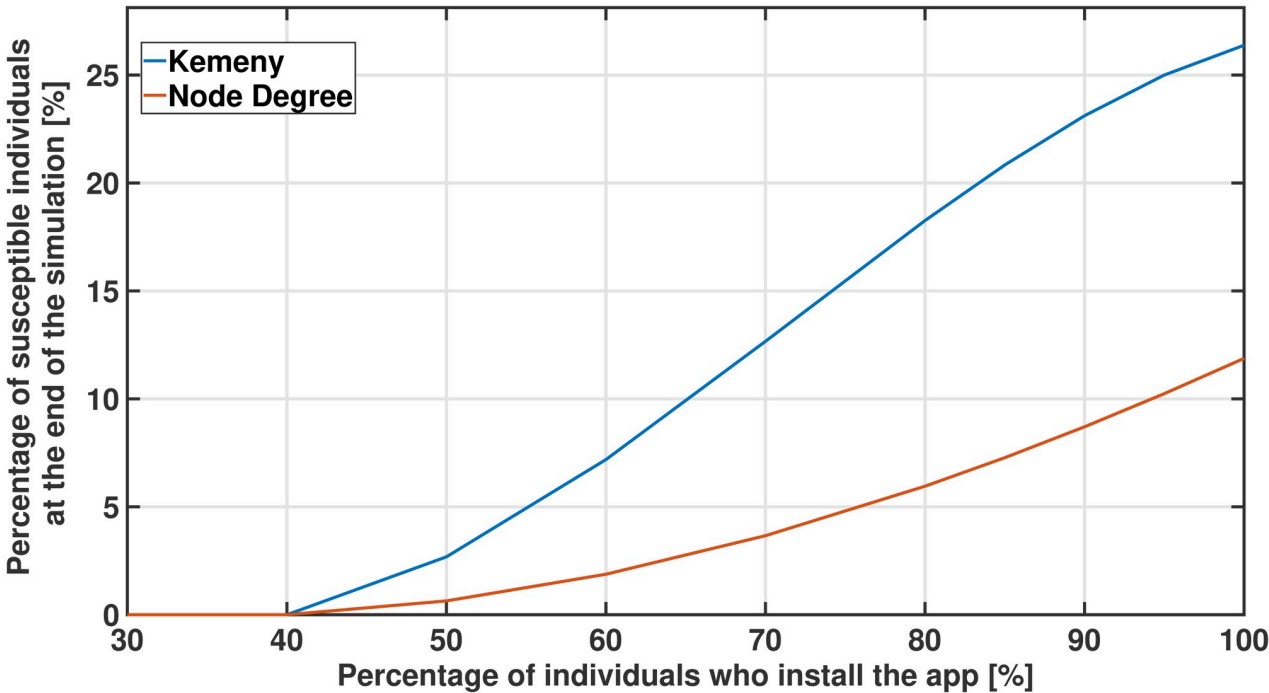

**Fig 8. Effect of compliance.** Percentage of individuals who remain healthy throughout the simulation of 30 days, assuming different percentages of individuals who install the contact tracing app. Kemeny-based testing appears more effective than node degree testing for all different levels of compliance.

less than 15 minutes (and are not recorded by the app), or even if a proper contact does not occur at all (for instance, if the infection is taken after touching an infected surface).

In this case, the interpretation of what people should be tested according to the Kemeny indicator is less intuitive, as not all edges are equally important (i.e., equally dangerous for spreading a virus), and the chosen individuals are not just bridges between communities. An example of what individuals are chosen is depicted in Fig 9, where the weights of edges are represented by changing the thickness of the edges (i.e., a thicker edge corresponding to a longer contact). The interest of this last section is that indicators that exploit transition matrices (e.g., like the Kemeny constant) may be used to analyze weighted graphs, while other indicators that exploit adjacency matrices (e.g., like the previously mentioned indicator based on random walk betweenness, that in the case of unweighted graphs provided similar results to the Kemeny constant) fail to take into account such important further information (e.g., duration of contacts).

## Discussion

While this is the first use of the Kemeny constant in sampling problems that we are aware of, it is related to several other methods in the literature. In particular, as mentioned, the connection to the random walk betweenness indicator is immediate. However, it is worth noting some important advantages of the Kemeny constant.

- First, from a merely computational point of view, calculation of the Kemeny constant is of the same computational complexity as that of calculating the determinant, and thus does scale well with the size of the network. From this perspective, it is more convenient than the random walk betweenness indicator.

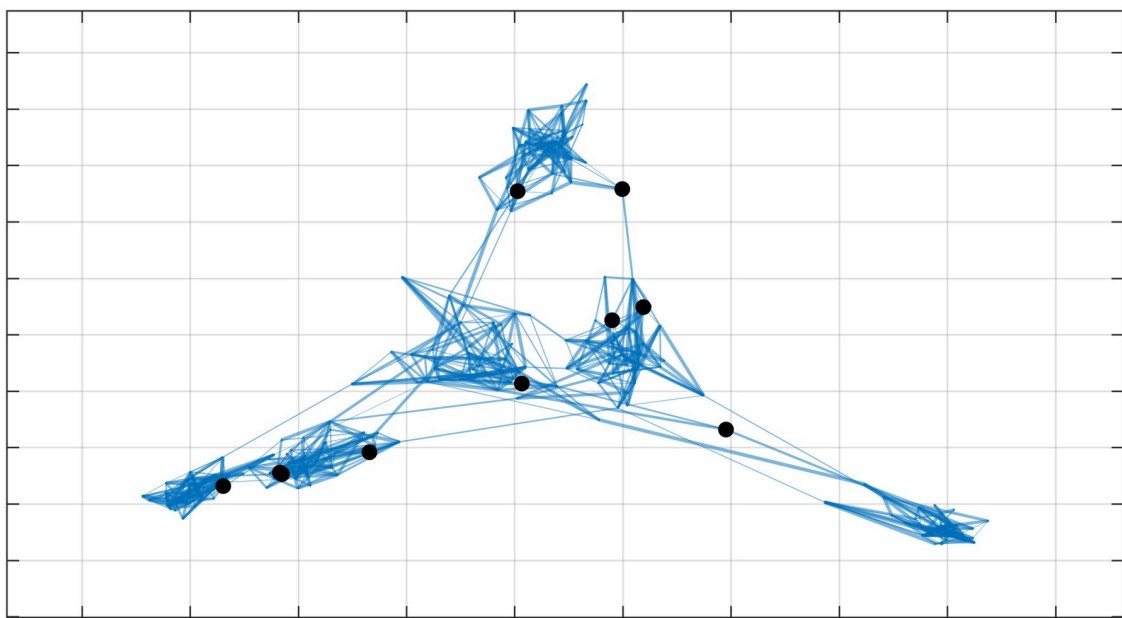

**Fig 9. Kemeny-testing in weighted graphs.** When edges have different weights (represented with lines of different thickness), then this further information is taken into account when choosing who should be tested according to the indicator based on the Kemeny constant.

- Second, differently from the random walk betweenness, and other similar indicators that are computed on the basis of the *adjacency matrix*, the Kemeny constant is computed using the *transition matrix*. Accordingly, it can take into account *weighted graphs*, as illustrated through a specific case study, where the probability of infections can be properly modelled, assuming that the contact tracing app has the ability to keep track of the duration of contacts, and/or of the relative distance. This allows for asymmetric and variable transmission probabilities between different communities [29].

- Third, *directionality* can be important. Diseases can have asymmetric transmission paths between compartments (e.g., in COVID-19 there is some evidence that children may be less susceptible to transmission than adults [30, 31] although in a study in Shenzhen, China, it was suggested that they were equally susceptible to infection [32]). Furthermore, behavioural differences between communities, such as non-homogeneous wearing of face-masks, is also likely to lead to further asymmetries in transmission (e.g., non wearers may be at an increased likelihood of infection, but are also much more likely to infect others). Such asymmetries lead to *directed graphs*. While the Kemeny constant readily extends to such situations, it is not immediately evident how other similar indicators extend to this case in a computationally efficient manner.

## Proactive intervention

Note that in most simulations, while the nodes remained in the same community throughout, the graph representing their interactions was drawn randomly each day, with further randomness about whether an edge was effective in transmitting disease. This means that individual nodes do not remain in their role as potential bridges between communities for more than that day. In the second simulation we employed a graph with fixed structure then simulated random interactions on the edges to provide a more realistic model which would have more

consistency between days, allowing us to benefit further from the precision of the Kemeny indicator.

In this scenario, nodes near bridges between communities would be likely to face repeated testing on multiple days. In real life this might face resistance, depending on the context. For example, while testing professionals, such as care workers repeatedly as part of their job would probably have high compliance, there may be others who would resist that. An alternative approach is to identify and intervene with bridge nodes proactively before infection. Such intervention could include quarantine measures, or more targeted education or police enforcement.

**Comment 1**: Our motivation is to make the research and public health communities aware of a tool that can be used in the context of contact tracing. Since most of the currently proposed contact tracing apps do not enable a central picture of the connectivity graph, a natural question concerns the utility of any such approach, given privacy concerns about centralised government knowledge of social graph structures. This is a valid concern. While not the concern of this present paper, we note here that distributed estimation of the Kemeny constant is in principle possible by initializing multiple random walks along the connectivity graph. Such strategies are considered in [33] in the context of a distributed reinforcement algorithm, and can be enabled and secured using distributed ledger technology. The study of such algorithms will be the subject of future publications.

**Comment 2**: In this manuscript we mainly focused on the consequences of node removal actions, as they correspond to quarantining an individual. However, one may think of some softer alternatives, such as warning individuals when they move from one community to another one. Given the shown dangerous potential of individuals who serve as bridges between different communities, it may be interesting to devise strategies to educate individuals (e.g., through smartphone warnings) to observe more prudential behaviours. While such warnings, on the one side, may be seen as privacy intrusive, and limiting of one's freedom, still they are less limiting than directly quarantining individuals, and also cheaper and quicker actions than testing.

**Comment 3**: This work directly applies the ideas pioneered in [13] by some of the authors of the present manuscript. In that paper the Kemeny indicator, had been used to identify critical nodes and edges in networks, according to a node removal and renormalization approach. Since publication of that paper in 2011, many authors have applied the same idea in different application domains; see for example [34] in Markov influence graphs.

## Conclusion

We have presented a framework for using the change in the Kemeny constant of a graph to identify and analyze bridges between sub-communities. The use of the Kemeny indicator is computationally convenient, and also supports the study of weighted and directed graphs.

Applications of testing, tracking and tracing will be critical in helping countries to safely reopen activities after the first wave of the COVID-19 virus, but they will be faced with limitations on the number of tests they can apply each day, and the compliance of the population in respecting quarantine isolation measures. The theoretical and simulation results presented in this paper show how the application of graph theory and the Kemeny indicator can be conveniently used to efficiently identify and block new virus outbreaks as early as possible by removing possible 'super-spreader' links that transmit disease between different communities.

The simulation models have deliberately been kept very simple, to illustrate the core concepts, but the work should be applicable to any simulation model which incorporates a graph or network representation of the population and their interactions. The work also has

implications for the design of tracking processes and apps, to ensure that they can provide complete information on both the adjacency of nodes and the transition matrix.

We believe that our work can be interestingly extended under a significant number of lines of research, including among others:

- As mentioned, we have not taken into account the dynamic evolution of contact graphs day after day. Similarly, information about who has been tested in the previous days could be further employed to decide who should be tested;

- So far, we have presented the Kemeny indicator method and the node degree as two alternative competing methods for deciding who should be tested. However, an optimized combination of individuals who maximize one, or the other, indicator may actually be the best solution to combine the advantages of the two methods (i.e., intercept 'super-spreaders' before they infect new communities, and to mitigate the spread within a community;

- While the Kemeny indicator appears to outperform high node degree also for the case of a lower compliance of installation of the tracing app, this remains an aspect of concern of the designed methodology. Accordingly, we believe that it could be very interesting to investigate approximated distributed approaches for computing the Kemeny constant. Also, for very large-scale networks, approximated and centralized solutions may be required for quick estimates of the Kemeny constant;

- While our paper so far aims to illustrate the differences between different indicators in terms of who should be tested in networks, more sophisticated epidemiological models may be used to better evaluate the impact of different testing strategies [35];

- Finally, our findings go beyond the specific application of interest that we analyze here: in fact, for instance, it may be of high interest for immunization procedures against other seasonal diseases as well.

## Acknowledgments

The authors thank Jane Breen, Ontario Tech University, Steve Kirkland, University of Manitoba, and Jakub Mareček, Czech Technical University in Prague, and all the reviewers for their valuable comments.

## Author Contributions

**Investigation:** Serife Yilmaz, Ekaterina Dudkina, Michelangelo Bin, Emanuele Crisostomi, Pietro Ferraro, Roderick Murray-Smith, Thomas Parisini, Lewi Stone, Robert Shorten.

**Methodology:** Serife Yilmaz, Ekaterina Dudkina, Michelangelo Bin, Emanuele Crisostomi, Pietro Ferraro, Roderick Murray-Smith, Thomas Parisini, Lewi Stone, Robert Shorten.

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
