## [Decision Letter · Decision Letter 0]

3 Oct 2020

PONE-D-20-23601

Kemeny-based testing for COVID-19

PLOS ONE

Dear Dr. Crisostomi,

Thank you for submitting your manuscript to PLOS ONE. After careful consideration, we feel that it has merit but does not fully meet PLOS ONE’s publication criteria as it currently stands. Therefore, we invite you to submit a revised version of the manuscript that addresses the points raised during the review process.

We look forward to receiving your revised manuscript.

Kind regards,

Ismael González Yero

Academic Editor

PLOS ONE

Journal Requirements:

2. Please note that PLOS ONE does not allow for the use of footnotes in its publications. As such, we ask you to remove all footnotes and move the information contained in them to the main text.

Reviewers' comments:

Reviewer's Responses to Questions

**Comments to the Author**

1. Is the manuscript technically sound, and do the data support the conclusions?

Reviewer #1: Yes

Reviewer #2: Yes

2. Has the statistical analysis been performed appropriately and rigorously? 

Reviewer #1: Yes

Reviewer #2: Yes

3. Have the authors made all data underlying the findings in their manuscript fully available?

Reviewer #1: Yes

Reviewer #2: Yes

4. Is the manuscript presented in an intelligible fashion and written in standard English?

Reviewer #1: Yes

Reviewer #2: Yes

5. Review Comments to the Author

Reviewer #1: This is a very good work.

It points out several important points in the analysis of the infection spread prevention in terms of testing. The paper is very well written solid and interesting. The first idea one can have of targeting the highly connected nodes can indeed be very misleading.

I definitely recommend publication. I do not have essential change to suggest.

Since I was very interested I add some minor comments/considerations. The Authors may decide to consider in the final version or in future work (or disregard them).

As a pure academic considerations, the proposed index can be referred to a group of nodes.

I mean there can be that two nodes (this is not quite realistic) that have very high Kemeny indicators, but if one of them is removed then the index of the other immediately drops. This is the case in which two communities are connected by two consecutive nodes, so once one of these is removed the other becomes

absolutely 'insignificant'. So if k nodes can be removed, then only one of these two

should be removed. Accordingly one can think to the k-index which is computed by considering the effect of removing k nodes. This leads clearly to a large complexity (combinatorial) in the evaluation.

Another important issue for the future is how one can compute, possibly in an approximate way, the suggested indicator for large network.

I would spend more words to the case of edge Kemeny index (achieved by removing an edge).

Finally, as an opinion, I think that these type of considerations will be more important in view of future vaccination strategies (when a vaccine will be available) rather than in testing.

MINOR

Kemeny constant tends to infinity

I would just say

Kemeny constant becomes infinite

The index Kemeny indicator is defined in a very colloquial way in the simulation section as follows:

For this purpose, we quantify the important of each single node as the corresponding value of the Kemeny constant of the graph obtained from the original graph by removing such a node, and for simplicity we shall denote it as the Kemeny indicator.

I would anticipate this as a more formal definition in the previous section.

Reviewer #2: This paper proposes an interesting method, based upon the so-called Kemeny constant, to determine which individuals to test, in case of limited resources for testing. The results are promising although the results are based on a limit number of simulations and relatively small networks.

Some detailed comments:

- It seems the order of the Figures doesn't correspond with the order of the caption in the text.

- Page 4: line 104: strange formulation; first it is mentioned that 1 is the spectral radius and then it says: 1 also belongs to the spectrum; this is redundant information

- page 6: line 176: I don't think the general reader knows what "tele-portation" means in the setting of PageRank. Also check spelling: teleportation?

-page 7: line 268: How is "meeting individuals from the same community" defined? Just related to connectivity or in terms of mobility or in terms of probability?

- page 8: line 237: "both solutions". Which solutions are you referring to?

- page 8: kine 249: "those based on random walks" Do you mean based upon the Kemeny indicator?

6. PLOS authors have the option to publish the peer review history of their article (what does this mean?). If published, this will include your full peer review and any attached files.

Reviewer #1: No

Reviewer #2: No

---

## [Author Response · Author response to Decision Letter 0]

19 Oct 2020

Revision of manuscript PONE-D-20-23601

We recommend both the EIC and the Reviewers to refer to the specific file (which should be found at the end of the manuscript) for the complete response to the comments of the Reviewers (where different colors and ad hoc figures can be appreciated). In the following, you will only find a simplified textual version of our responses.

Dear Editor-in-Chief,

Please find enclosed the revision of manuscript PONE-D-20-23601 “Kemeny-based testing for COVID-19”.

Our responses to the comments of the Reviewers have been incorporated into the revised manuscript. Also, point by point responses to the comments are reported below (Reviewers’ comments are reported in black, while our responses are in blue).

We hope that the paper is a significant improvement on the previous one and can now be accepted for publication.

Yours sincerely,

Emanuele Crisostomi

(on behalf of my co-authors)

 

Reviewer #1: This is a very good work.

It points out several important points in the analysis of the infection spread prevention in terms of testing. The paper is very well written solid and interesting. The first idea one can have of targeting the highly connected nodes can indeed be very misleading.

I definitely recommend publication. I do not have essential change to suggest. Since I was very interested I add some minor comments/considerations. The Authors may decide to consider in the final version or in future work (or disregard them).

We thank the Reviewer for the general very positive feedback. Below our responses to the specific comments.

As a pure academic considerations, the proposed index can be referred to a group of nodes.

I mean there can be that two nodes (this is not quite realistic) that have very high Kemeny indicators, but if one of them is removed then the index of the other immediately drops. This is the case in which two communities are connected by two consecutive nodes, so once one of these is removed the other becomes absolutely 'insignificant'. So if k nodes can be removed, then only one of these two should be removed. Accordingly one can think to the k-index which is computed by considering the effect of removing k nodes. This leads clearly to a large complexity (combinatorial) in the evaluation.

The Reviewer is raising an interesting point. It is possible to devise “ad hoc” rules to improve the basic functioning of the Kemeny-based indicator to address some possible limit cases (one being the example provided by the Reviewer). For instance, one possible improvement (which, if we understand correctly, is different from the one suggested by the Reviewer) is to choose the single node with highest ranking; remove it from the network; and then rank all nodes again according to the new network (and repeat this procedure until the list of individuals who should be tested is complete). 

Such a procedure would be more computationally demanding, but would actually improve not only the ranking of the Kemeny-based indicator, but of all the other indicators as well. 

In our work, we decided to simultaneously compute the most critical nodes according to all indicators, to maintain the presentation and the computations simple, while preserving the fairness of the comparison. 

In the revised manuscript we have added a remark to clarify that an alternative way of choosing the most critical nodes could be adopted.

Another important issue for the future is how one can compute, possibly in an approximate way, the suggested indicator for large network.

This is also an interesting point. According to the final application of our proposed methodology, one may be interested in obtaining the solution in a very short time. Since (one way of) computing the Kemeny constant requires the computation of all the eigenvalues of the transition matrix (see Equation (4)), a straightforward approximation of the indicator may only involve the second largest eigenvalue modulus (SLEM) (or the first few eigenvalues for a more accurate approximation). 

We however prefer to leave this problem as a future work, together with the somewhat related problem of obtaining a distributed estimate of the indicator. 

I would spend more words to the case of edge Kemeny index (achieved by removing an edge).

We assume that here the Reviewer was actually referring to the Kemeny index achieved by removing a node (and not an edge). In this case, in the revised manuscript we now anticipate the more formal definition to the previous section, as suggested by the same Reviewer in a later question.

If the Reviewer was truly referring to the case of the Kemeny constant after removing an edge, we refer to an earlier published paper of some of the authors where the problem of estimating the change of the Kemeny constant by varying some non-diagonal entries of the transition matrix is address [1, Equation (14)]. However, while node removal makes perfect sense in the application of interest here (as it corresponds to quarantine), edge removal has a less straightforward interpretation here (prevent a single specific contact between two individuals from occurring?), and the theory developed in [1] has not been used in the current work.

[1] E. Crisostomi, S. Kirkland and R. Shorten, “A Google-like model of road network dynamics and its application to regulation and control”, International Journal of Control, vol. 84, no. 3, pp. 633-631.

Finally, as an opinion, I think that these type of considerations will be more important in view of future vaccination strategies (when a vaccine will be available) rather than in testing.

We agree with the Reviewer that the methodology can be perfectly used for vaccination strategies in general, independently also from the specific COVID-19 case.

MINOR

- Kemeny constant tends to infinity I would just say Kemeny constant becomes infinite

The Kemeny constant had been originally defined for fully connected graphs, so, in principle, some mathematicians may argue that it should never occur that it may become infinite. That is why we tried to mitigate the sentence that “tends” to infinity.

- The index Kemeny indicator is defined in a very colloquial way in the simulation section as follows: “For this purpose, we quantify the important of each single node as the corresponding value of the Kemeny constant of the graph obtained from the original graph by removing such a node, and for simplicity we shall denote it as the Kemeny indicator”. I would anticipate this as a more formal definition in the previous section.

As from an earlier comment of the same Reviewer, we have now revised this aspect as suggested.

Reviewer #2: This paper proposes an interesting method, based upon the so-called Kemeny constant, to determine which individuals to test, in case of limited resources for testing. The results are promising although the results are based on a limit number of simulations and relatively small networks.

Some words about this first comment can be found in the response letter (and are not reported here since figures can not be uploaded in this space).

Some detailed comments:

- It seems the order of the Figures doesn't correspond with the order of the caption in the text.

In the revised submission, we have paid attention in ordering the figures in the same way of the caption in the text.

- Page 4: line 104: strange formulation; first it is mentioned that 1 is the spectral radius and then it says: 1 also belongs to the spectrum; this is redundant information

In the revised manuscript we have changed this into “The spectral radius of P is 1; in particular, 1 belongs to the spectrum of P”. However, note that there is no redundancy: the first part does not imply the second one, as it could be that only -1 (and not 1) was in the spectrum of P. Also, the second part does not imply the first one, as if 1 belongs to the spectrum, it does not imply that 1 is also the eigenvalue of largest modulus. 

- page 6: line 176: I don't think the general reader knows what "tele-portation" means in the setting of PageRank. Also check spelling: teleportation?

We thank the Reviewer for this comment. In the revised version of the manuscript we have clarified the meaning of teleportation, and we have adopted the recommended spelling without the hyphen symbol.

-page 7: line 268: How is "meeting individuals from the same community" defined? Just related to connectivity or in terms of mobility or in terms of probability?

It is in terms of probability. We now briefly explain how: we first associate each individual with a community. Then, we add edges between nodes (i.e., individuals) with a certain probability: the probability that we add an edge between two individuals of the same community is 19%, while the probability that we add an edge between two individuals belonging to two different communities is 0.1%. Every day (apart from the simulations based on the same fixed scenario), a new (and in principle completely different) graph is generated according to this same procedure.

Obviously, the information used to create the graph (e.g., the community of each individual) is then not exploited by any of the indicators used to decide who should be tested. 

- page 8: line 237: "both solutions". Which solutions are you referring to?

We were referring to the possibility of testing individuals either on the basis of the ranking of node degree, or of the proposed Kemeny-based indicator. In the revised version of the manuscript, we have revised this sentence to clarify our meaning. 

- page 8: kine 249: "those based on random walks" Do you mean based upon the Kemeny indicator?

We were meaning both the proposed Kemeny-based indicator, and also the “Random Walk Betweenness” indicator proposed by Newman in reference [23]. In the revised version of the manuscript, we have revised this sentence to clarify our meaning.

---

## [Decision Letter · Decision Letter 1]

3 Nov 2020

Kemeny-based testing for COVID-19

PONE-D-20-23601R1

Dear Dr. Crisostomi,

We’re pleased to inform you that your manuscript has been judged scientifically suitable for publication and will be formally accepted for publication once it meets all outstanding technical requirements.

Kind regards,

Ivan Kryven

Academic Editor

PLOS ONE

Additional Editor Comments (optional):

Reviewers' comments:

Reviewer's Responses to Questions

**Comments to the Author**

1. If the authors have adequately addressed your comments raised in a previous round of review and you feel that this manuscript is now acceptable for publication, you may indicate that here to bypass the “Comments to the Author” section, enter your conflict of interest statement in the “Confidential to Editor” section, and submit your "Accept" recommendation.

Reviewer #1: All comments have been addressed

Reviewer #2: All comments have been addressed

2. Is the manuscript technically sound, and do the data support the conclusions?

Reviewer #1: Yes

Reviewer #2: (No Response)

3. Has the statistical analysis been performed appropriately and rigorously? 

Reviewer #1: N/A

Reviewer #2: (No Response)

4. Have the authors made all data underlying the findings in their manuscript fully available?

Reviewer #1: Yes

Reviewer #2: (No Response)

5. Is the manuscript presented in an intelligible fashion and written in standard English?

Reviewer #1: Yes

Reviewer #2: (No Response)

6. Review Comments to the Author

Reviewer #1: (No Response)

Reviewer #2: (No Response)

7. PLOS authors have the option to publish the peer review history of their article (what does this mean?). If published, this will include your full peer review and any attached files.

Reviewer #1: No

Reviewer #2: No

---

## [Editor Report · Acceptance letter]

10 Nov 2020

PONE-D-20-23601R1 

Kemeny-based testing for COVID-19 

Dear Dr. Crisostomi:

I'm pleased to inform you that your manuscript has been deemed suitable for publication in PLOS ONE. Congratulations! Your manuscript is now with our production department. 

Kind regards, 

on behalf of

Dr. Ivan Kryven 

Academic Editor

PLOS ONE